

# Augmented cartilage regeneration by implantation of cellular versus acellular implants after bone marrow stimulation: a systematic review and meta-analysis of animal studies

Michiel W. Pot[1], Toin H. van Kuppevelt[1], Veronica K. Gonzales[2], Pieter Buma[2], Joanna IntHout[3], Rob B.M. de Vries[4] and Willeke F. Daamen[1]

[1] Department of Biochemistry, Radboud Institute for Molecular Life Sciences, Radboud university medical center, Nijmegen, The Netherlands
[2] Department of Orthopedics, Radboud Institute for Molecular Life Sciences, Radboud university medical center, Nijmegen, The Netherlands
[3] Department for Health Evidence, Radboud Institute for Health Sciences, Radboud university medical center, Nijmegen, The Netherlands
[4] SYRCLE (SYstematic Review Centre for Laboratory animal Experimentation), Central Animal Laboratory, Radboud university medical center, Nijmegen, The Netherlands

Corresponding authors
Michiel W. Pot,
Michiel.Pot@radboudumc.nl
Willeke F. Daamen,
Willeke.Daamen@radboudumc.nl

## ABSTRACT

Bone marrow stimulation may be applied to regenerate focal cartilage defects, but generally results in transient clinical improvement and formation of fibrocartilage rather than hyaline cartilage. Tissue engineering and regenerative medicine strive to develop new solutions to regenerate hyaline cartilage tissue. This systematic review and meta-analysis provides a comprehensive overview of current literature and assesses the efficacy of articular cartilage regeneration by implantation of cell-laden versus cell-free biomaterials in the knee and ankle joint in animals after bone marrow stimulation. PubMed and EMBASE (via OvidSP) were systematically searched using tissue engineering, cartilage and animals search strategies. Included were primary studies in which cellular and acellular biomaterials were implanted after applying bone marrow stimulation in the knee or ankle joint in healthy animals. Study characteristics were tabulated and outcome data were collected for meta-analysis for studies applying semi-quantitative histology as outcome measure (117 studies). Cartilage regeneration was expressed on an absolute 0–100% scale and random effects meta-analyses were performed. Implantation of cellular biomaterials significantly improved cartilage regeneration by 18.6% compared to acellular biomaterials. No significant differences were found between biomaterials loaded with stem cells and those loaded with somatic cells. Culture conditions of cells did not affect cartilage regeneration. Cartilage formation was reduced with adipose-derived stem cells compared to other cell types, but still improved compared to acellular scaffolds. Assessment of the risk of bias was impaired due to incomplete reporting for most studies. Implantation of cellular biomaterials improves cartilage regeneration compared to acellular biomaterials.

## INTRODUCTION

Articular cartilage facilitates joint loading and movement by resisting compressive and shear forces (*Swieszkowski et al., 2007*). For patients, localized cartilage defects can have detrimental long term effects such as joint dysfunction, pain, and degenerative osteoarthritis. Upon cartilage damage, its avascular nature prevents spontaneous healing (*Buckwalter, Saltzman & Brown, 2004*). Clinical treatments for full-thickness cartilage defects and osteochondral lesions include bone marrow stimulation techniques, e.g., microfracture and subchondral drilling, and autologous chondrocyte implantation. Defect size generally determines treatment, where microfracture and autologous chondrocyte implantation are used to treat small ($<2.5$ cm$^2$) and large lesions ($>2.5$ cm$^2$), respectively (*Cucchiarini et al., 2014*). Microfracture surgery is a minimally invasive and inexpensive one-step approach, where multiple perforations, microfractures, are made in the subchondral bone plate to induce bleeding and provoke a reparative response. The formed blood clot consists of bone marrow-derived mesenchymal stem cells (BM-MSCs), growth factors and other proteins, supporting cartilage formation (*Steadman, Rodkey & Rodrigo, 2001*). The repaired tissue, however, generally consists of fibrous cartilage, which lacks the mechanical properties of native hyaline cartilage (*Dai et al., 2014*). Microfracture results in temporary clinical improvement only (*Saris et al., 2014*), and the demand for improved cartilage regeneration persists.

Cartilage regeneration may be improved by tissue engineering and regenerative medicine (TERM) in addition to bone marrow stimulating techniques. TERM encompasses the development of biomaterials, which can be loaded with cells and biologics (*Seo et al., 2014*). Upon implantation and infiltration of BM-MSCs, the biomaterial may act as a template to guide/stimulate cartilage regeneration (*Cucchiarini et al., 2014*). In a previous systematic review and meta-analysis on animal models, we showed that acellular biomaterials in addition to bone marrow stimulation was more effective in regenerating cartilage *in vivo* than bone marrow stimulation alone, which was further improved by use of biologics (*Pot et al., 2016*).

When biomaterials are loaded with cells, bone marrow stimulation may be even more effective. Biomaterials loaded with cells after bone marrow stimulation has been widely investigated *in vivo*, and included loading of chondrocytes (*Ahn et al., 2009*; *Caminal et al., 2016*; *Christensen et al., 2012*), BM-MSCs (*Araki et al., 2015*; *Igarashi et al., 2012*; *Wakitani et al., 1994*), synovium-derived mesenchymal stem cells (SD-MSCs) (*Pei et al., 2009*; *Lee et al., 2013*; *Shimomura et al., 2014*), adipose-derived stem cells (ADSCs) (*Xie et al., 2012*; *Masuoka et al., 2006*; *Kang et al., 2014*), periosteal cells (*Perka et al., 2000*; *Schagemann et al., 2009*), fibroblasts (*Yan & Yu, 2007*), umbilical cord stem cells (UCSC) (*Yan & Yu, 2007*; *Chung et al., 2014*) and embryonic stem cells (ESC) (*Cheng et al., 2014*). Cells are either used directly after harvesting (*Betsch et al., 2014*; *Getgood et al., 2012*) or after an additional *in vitro* step of cell expansion (*Guo et al., 2010*; *Dorotka et al., 2005*) and/or differentiation (*Sosio et al., 2015*; *Necas et al., 2010*).

In this systematic review and meta-analysis, we present a comprehensive overview of all current literature regarding regeneration of articular cartilage by implantation of cell-laden versus cell-free biomaterials in the knee and ankle joint after bone marrow stimulation in animal models (Fig. 1). We further investigated the effect of loading biomaterials with (1) stem

**Cartilage regeneration by implantation of acellular or cell-laden biomaterials**

**Figure 1** Illustration of articular cartilage regeneration by implantation of cellular and acellular biomaterials after applying bone marrow stimulation. The figure was adapted from *Pot et al. (2016)*.

cells versus somatic (differentiated) cells, (2) different cell types (e.g., chondrocytes, MSCs, ADSCs), and (3) culture conditions of cells (e.g., use after harvesting, *in vitro* expansion and/or differentiation). In the meta-analysis, histological scores from semi-quantitative histological scoring systems were used to assess the effect on cartilage regeneration.

# MATERIALS AND METHODS

## Search strategy

An extensive literature search was performed in PubMed and EMBASE (via OvidSP) to identify relevant peer-reviewed articles until June 29, 2016, using methods defined by *De Vries et al. (2012)* and *Leenaars et al. (2012)*. The search strategy

(Supplemental Information 1) consisted of search components for tissue engineering (*Sloff et al., 2014*) and cartilage (*Pot et al., 2016*). Results were refined for animal studies by applying animal search filters (*Hooijmans et al., 2010*; *De Vries et al., 2011*). No language restrictions were applied.

## Study selection

After obtaining all references, duplicates were manually removed in EndNote X7 (Thomson Reuters, Philadelphia, PA, USA) by one author (MP). Resulting references were screened for relevance by two independent authors (MP and VG/WD) based on title, title/abstract and full-text using Early Review Organizing Software (EROS, Institute of Clinical Effectiveness and Health Policy, Buenos Aires, Argentina, http://www.eros-systematic-review.org). In case of disagreement between authors or any doubt, references were included for further screening. An overview of all exclusion criteria per screening phase is provided in Supplemental Information 2.

Studies were included for risk of bias assessment and meta-analysis when semi-quantitative histological scoring was used as outcome measure.

## Study characteristics

Study characteristics were extracted from the studies by MP. Basic information (author, year of publication), animal model characteristics (species, strain, sex, etc.), experimental characteristics (surgery, biomaterial, follow-up, etc.), cell characteristics (cell type, culture conditions, etc.) and outcome characteristics (macroscopic evaluation, histology and semi-quantitative histological scoring, etc.) were obtained.

## Risk of bias assessment

The methodological quality was assessed for studies included in the meta-analysis. A risk of bias analysis was performed according to an adapted version (*Pot et al., 2016*) of the tool described by *Hooijmans et al. (2014)*. Selection, performance, detection and attrition bias were scored independently by MP and VG/WD using questions and a flowchart (*Pot et al., 2016*), where '-', '?' and '+', indicating low, unknown and high risk of bias. In case of differences between authors, results were discussed until consensus was reached. Unfortunately, 16 articles were published in Chinese and we did not have the resources to obtain certified translations of these articles. We were, however, able to successfully extract the data of these studies using Google Translate (https://translate.google.com/) and used the data in the meta-analysis. A sensitivity analysis was performed to evaluate the effect of language (exclusion of Chinese articles, see 'Meta-analysis').

## Analysis preparations and meta-analysis
### Analysis preparations

Meta-analyses were performed for outcome measure semi-quantitative histology; data were used from studies that compared biomaterials with (experimental group) and without cells (control group). In general, these histological scoring systems and their components, extensively reviewed by *Rutgers et al. (2010)*, evaluate the degree of cartilage regeneration by scoring parameters like Safranin-O staining (which stains negatively

charged glycosaminoglycans, an important component of cartilage tissue), surface integrity and cartilage thickness.

Outcome data (mean, standard deviation (SD) and number of animals) were extracted from the studies for all time points as follows: (1) numerical data from the text/tables, (2) graphical results by measuring the mean and SD using ImageJ (1.46r, National Institutes of Health USA), (3), boxplot results by recalculating from median, range and sample size to mean and SD (*Hozo, Djulbegovic & Hozo, 2005*), and (4) for results presented as mean and confidence interval (CI) per group, the following equation was used to recalculate CI to a standard deviation: $SD = \sqrt{N} \times \frac{upper\ limit - lower\ limit}{3.92}$ for a 95% CI (*Higgins & Green, 2011*). When data were missing or unclear, authors were contacted to provide data. Studies were excluded from meta-analysis in case data could not be retrieved or remained unclear (e.g., missing SD, all SD's similar to corresponding mean, and histological scores exceeding maximum), unless data were sufficiently clear to make assumptions (i.e., group size and number of animals per time point and analyses, see Supplemental Information 3). A sensitivity analysis was performed to evaluate the effect of assumptions (exclusion of articles with assumptions, see 'Meta-analysis'). Histological scoring systems describe the degree of cartilage regeneration with different scoring scales. To compare data from different studies, all data were converted to a 100% cartilage regeneration scale by dividing both the mean and SD by the maximum score of the scoring system and multiplying the outcome by 100%. In this systematic review, healthy tissue is represented as 100% cartilage regeneration. Lower percentages indicate less regenerated cartilage. When results of experimental groups could be combined per study (i.e., outcome of various biomaterials seeded with one cell type), we did so, followed the approach described in the Cochrane Handbook, table 7.7 (*Higgins & Green, 2011*), which means that we calculated a weighted average of the results with an appropriate standard deviation. Time points of treatment groups were combined using the same approach. The mean and corresponding standard error (SE) per treatment group were subsequently calculated per study.

### Meta-analysis

The main research question was: Is there an overall beneficial effect on cartilage regeneration of implanting biomaterials loaded with cells compared to acellular biomaterials?

We used a bivariate approach to model a random effects meta-analysis, i.e., separate outcomes for the control and experimental group were used with their respective SEs. The correlation between these two outcomes was modeled with a compound symmetry covariance matrix, as this resulted in a lower Akaike Information Criterion value than an unstructured covariance matrix.

To evaluate the effect of specific variables on treatment outcome for the experimental group (biomaterials loaded with cells), the following sub-questions were addressed: (1) Is there a difference between the use of stem cells and somatic (differentiated) cells (stem cells vs. somatic cells); (2) Do differences among various cell subgroups exist (e.g., chondrocytes vs. other cells); (3) Is there a difference between biomaterials loaded with cells which were not cultured *in vitro*, were expanded *in vitro* or were differentiated *in vitro* (during surgery vs. expansion, surgery vs. differentiation, and expansion vs. differentiation)? Results are

depicted as % cartilage regeneration (95% CI: [lower CI, upper CI]. The mean difference (% [95% CI]) is presented as condition A–condition B. Based on a previous study, data of all time points were used (*Pot et al., 2016*). Subgroup analyses were performed in case subgroups consisted of more than five experimental groups in at least three studies. Most studies contained more than one experimental group, therefore the total number of studies and number of experimental groups (no. of studies/groups) is provided in the analysis. No adjustment for multiple testing was applied in analyses of sub-questions.

Sensitivity analyses were performed on the main research question to evaluate the effect of language (excluding Chinese articles, as the risk of bias for these articles was not investigated), and the effect of assumptions (excluding articles for which assumptions were made) in the meta-analysis.

SAS/STAT® software version 9.2 for Windows, copyright© 2002–2008 by SAS Institute Inc., Cary, NC, USA, was used to perform statistical analyses. R software version 3.0.1 (*R Core Team, 2011*) with package meta (*Schwarzer, 2015*) was used to create the funnel plot, which illustrates effect sizes of all studies versus their precision, and test for the asymmetry, using the method of moments estimator for the between study variation (*Thompson & Sharp, 1999*). $I^2$ was used as a measure of heterogeneity. $I^2$ measures the percentage of variability in treatment effect estimates that is due to between study heterogeneity rather than chance (*Higgins et al., 2003*). If $I^2$ is 0%, this suggests that the variability in the study estimates is entirely due to chance. If $I^2$ is >0% there might be other reasons for variability. ReviewManager (*The Cochrane Collaboration, 2014*) was used to create the forest plot.

## RESULTS

### Search and study inclusion

Searching PubMed and EMBASE databases for references regarding cartilage regeneration by implantation of cellular and acellular biomaterials in the knee and ankle joint in combination with bone marrow stimulation resulted in a total of 11,248 references (Pubmed 4,743, Embase 6,505). Removal of duplicates left 7,354 references. Screening by title and title/abstract resulted in exclusion of 6,744 references. Full-text of 610 studies resulted in 146 included studies. The full-text of some studies (*Xie et al., 2014*; *Yao, Ma & Zhang, 2000*; *Zhou & Yu, 2014*) could not be retrieved and these were excluded.

In the meta-analysis, studies were used which applied semi-quantitative histology as outcome measure, resulting in 117 included studies. A risk of bias assessment (Fig. 2) was performed for 101 of 117 studies (excluding Chinese studies). Supplemental Information 3 provides an overview of all included studies after full-text screening, risk of bias assessment and meta-analysis, as well as detailed information regarding reasons for exclusion and assumptions made for certain studies. Supplemental Information 4 contains the reference list and abbreviations of Supplemental Information 3 studies.

### Study characteristics

A large variation between studies was observed regarding animal model characteristics (species, strain, sex, etc.), experimental characteristics (surgery, biomaterial, follow-up, etc.), cell characteristics (cell type, culture conditions, etc.) and outcome characteristics
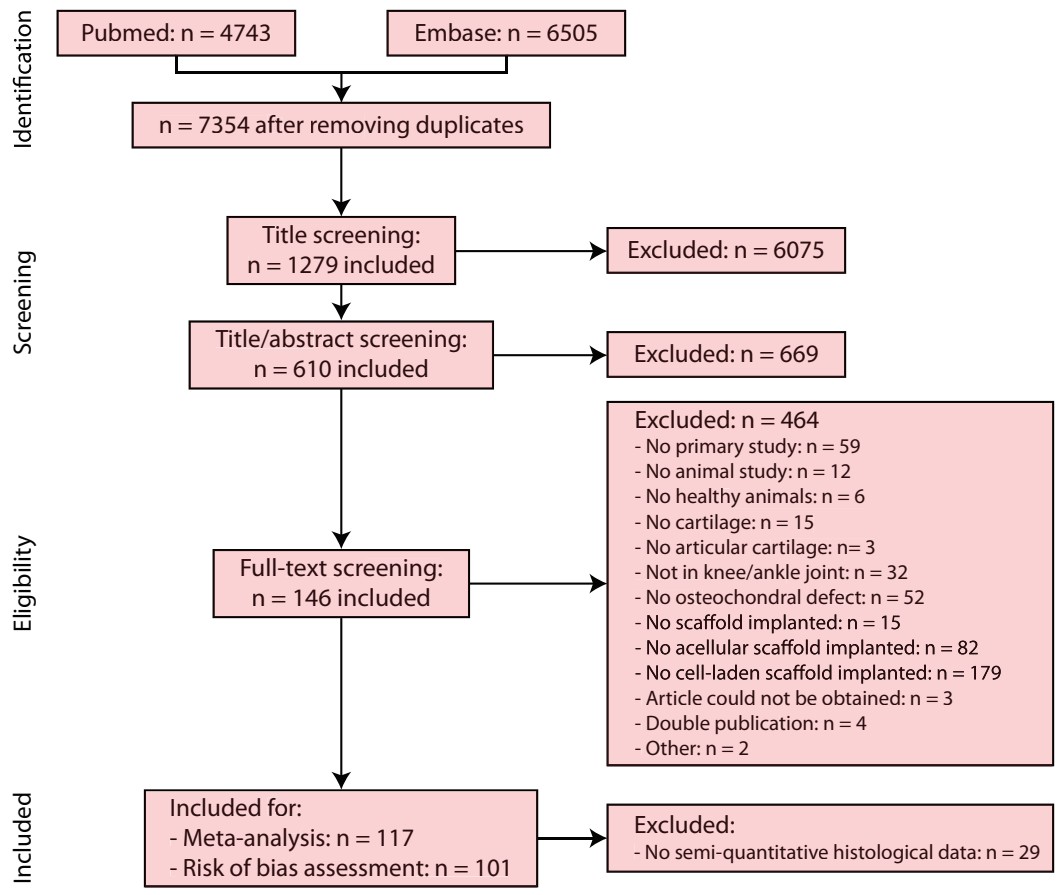

**Figure 2  PRISMA (Preferred Reporting Items for Systematic Reviews and Meta-analysis) flowchart of the systematic search of literature.** Of the 117 studies included for the meta-analysis, a risk of bias assessment was performed for 101 studies, excluding Chinese articles.

(macroscopic evaluation, histology and semi-quantitative histological scoring, etc.), as can be appreciated from Supplemental Information 3. Various animal species were used including rabbit, dog, sheep, pig, rat, horse, minipig, goat and macaques. A large range was found in animal age, e.g., the age of rabbits ranged from six weeks to >2 years. Small animals were generally younger (in the range of months) compared to larger animals (in the range of years). In many studies, no detailed information was provided regarding the animal's absolute age, but merely e.g., adult or mature.

The method for bone marrow stimulation was mostly subchondral drilling (142 studies), where only four studies used microfracture. Defects were created at various locations (trochlea, condyles, femur and intercondylar fossa) and with diverse dimensions (e.g., for rabbits: diameter 4–7 mm and depth 0.8–9 mm).

Implanted biomaterials were prepared from natural (e.g., alginate and collagen), synthetic (e.g., poly(lactic-coglycolic acid) and polycaprolactone) or mixtures of natural and synthetic materials. In 27 studies biologics, such as bone morphogenetic protein 2 and transforming growth factor beta, were loaded in the biomaterials. Different cell

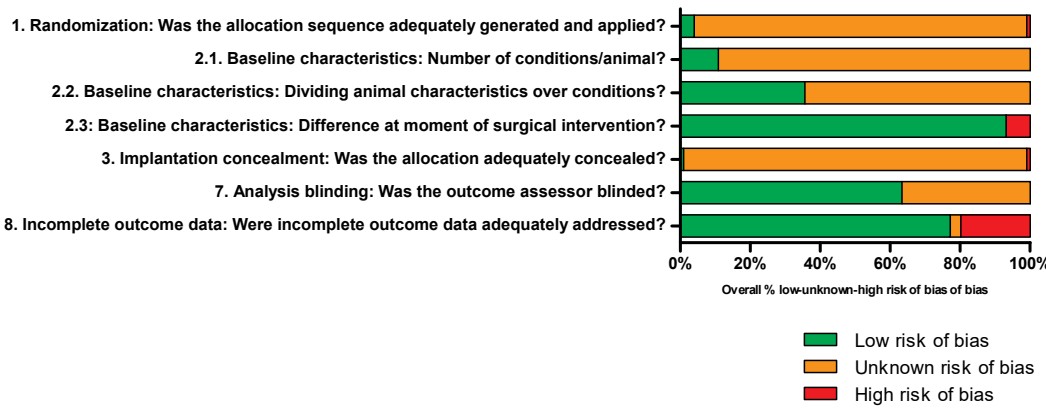

**Figure 3** **Results of the risk of bias analysis.** Low, unknown or high risk of bias are presented in green, orange and red, respectively, where the percentages indicate the percentage of studies scoring low, unknown or high risk of bias of the total number of investigated studies per question. Low risk of bias was mainly found for addressing incomplete outcome data and baseline characteristics at the moment of surgical intervention. Unknown risk of bias was generally the result of limited details described in the studies regarding the experimental set-up. High risk of bias was only occasionally scored. Questions 4–6 are not depicted graphically, but are described and explained in Supplemental Information 4.

types were applied, including chondrocytes, bone marrow-derived mesenchymal stem cells (BM-MSCs), bone marrow-derived progenitor cells, synovium-derived stem cells (SD-MSCs), bone marrow-derived mononuclear cells, adipose-derived stem cells, adipose-derived stromal vascular fraction cells, endothelial progenitor cells, embryonic stem cells, umbilical cord blood stem cells, fibroblasts, and periosteal cells, while in some studies undefined cell populations like bone marrow aspirate concentrate were used. Cells were either seeded directly after harvesting on biomaterials and implanted in the created defect or cultured *in vitro* to expand and/or differentiate the cells, followed by seeding on biomaterials and implantation. *In vitro* differentiation was performed with cells cultured in monolayer (without biomaterials), followed by seeding of the cells onto the biomaterials and implantations, or by directly culturing the cells on biomaterials prior to implantation.

In most studies, short-term cartilage regeneration was investigated: the follow-up time was generally less than 6 months with a maximum follow-up of 12 months.

## Risk of bias assessment

The methodological quality was assessed for all studies included in the meta-analysis except Chinese articles. The overview of the results in Fig. 3 indicates a general lack of information regarding the experimental setup of the studies, limiting the assessment of the actual risk of bias. Please see Supplemental Information 5 for all scores per individual study.

In the assessed studies, details regarding the application and method of randomization (Q1) were generally lacking. As a result, assessment of the actual risk of selection bias was practically impossible. Assessment of the actual risk of bias due to differences in baseline characteristics was difficult since no details regarding randomization were described. Differences may have been present in load-bearing between implantation sites (Q2.1) and age, sex and weight of animals (Q2.2). In most studies, few differences were found between

animals at the moment of surgical intervention since animals were treated similarly (Q2.3). Details regarding blinding of experimental conditions at the moment of implantation were generally not provided, which may have resulted in bias (Q3). Random housing of animals was generally not (well) described (Q4). Caregivers and/or investigators did not know which intervention each animal received during the experiment (Q5). No details were presented regarding the random selection of animals for outcome assessment (Q6). The method of blinding during analysis, however, was well described in most studies (Q7). Incomplete outcome data were identified or described in a few studies only, which resulted in studies with high risk of bias (Q8). Generally, most studies lacked reporting of important details and therefore adequate assessment of the actual risk of bias was difficult.

## Data synthesis

Semi-quantitative histological scores were used as outcome data to compare biomaterials with cells (experimental group) and without cells (control group) and to address sub-questions related to the use of type of cells and culture conditions. An overview of all meta-analysis results is provided in Table 1; an overview of all raw data is given in Supplemental Information 6.

Data are presented as the effect (%) with 95% CI, where 100% cartilage regeneration represents healthy tissue and lower percentages indicate less regenerated cartilage tissue.

### Overall effect implantation of cellular and acellular biomaterials

The meta-analysis indicates that implantation of cellular and acellular biomaterials resulted in 61.5% (95% CI [58.5–64.5]) and 43.0% (95% CI [40.0–46.0]) cartilage regeneration, respectively. The addition of cells to biomaterials significantly improved cartilage regeneration by 18.6% (95% CI [15.2–22.0], $p < 0.0001$). An overview of results for each individual study is displayed in the forest plot (Supplemental Information 7), presenting improved cartilage regeneration by loading biomaterials with cells in 66 studies, similar cartilage regeneration in 30 studies, and a negative effect on cartilage regeneration in two studies. The heterogeneity ($I^2$) for the comparison between cellular and acellular biomaterials was very high (99.4% (95% CI [99.3%–99.4%])).

### Stem cells and somatic cells

No significant differences ($p = 0.622$) were found between biomaterials loaded with stem cells (61.5% (95% CI [58.1–65.0])) and somatic cells (62.8% (95% CI [58.5–67.1])).

### Cell type

Biomaterials were loaded with various cell types. Subgroup analyses were only performed when subgroups consisted of more than five experimental groups in at least 3 studies. Seeding biomaterials with adipose-derived stem cells significantly decreased cartilage regeneration, while no other significant differences were observed (Table 1). Only for scaffolds seeded with adipose-derived stem cells (ADSCs), reduced cartilage regeneration was found (56.3% (95% CI [49.9–62.6])) compared to cellular scaffolds. However, cartilage regeneration using ADSCs-seeded scaffolds still improved regeneration compared to acellular scaffolds.

**Table 1 Overview meta-analysis results; the effect on cartilage regeneration of (1) the addition of cells to biomaterials, (2) loading of stem cells vs. somatic cells, (3) loading of specific cell types, e.g., chondrocytes vs. all cells except chondrocytes, and (4) culture conditions.** The total number of studies and number of groups included in the meta-analysis are depicted (studies may have > 1 experimental group, no. of studies/groups). Results are presented on a 100% cartilage regeneration scale, where 100% indicates 'maximum' cartilage regeneration. The addition of cells to biomaterials significantly improved cartilage regeneration compared to acellular biomaterials. The use of stem cells or somatic cells resulted in comparable cartilage regeneration. Cartilage regeneration was significantly lower for biomaterials seeded with adipose-derived stem cells compared to other cell types. Cartilage regeneration was not affected by the method of cell manipulation.

| Meta-analysis | No. of studies/groups | Subgroups | Cartilage regeneration (% [95% CI]) | Mean difference (% [95% CI]) *p*-value |
|---|---|---|---|---|
| 1. Overall effect | 98/265 | Cellular scaffolds | 61.5 [58.5–64.5] | 18.6% [15.2–22.0] |
| | 98/208 | Acellular scaffolds | 43.0 [40.0–46.0] | ***p* < 0.0001** |
| 2. Stem cells or somatic cells | 57/148 | Stem cells | 61.5 [58.1–65.0] | −1.28 [−6.5–4.0] |
| | 36/101 | Somatic cells | 62.8 [58.5–67.1] | *p* = 0.622 |
| 3. Type of cells | 30/81 | Chondrocytes | 63.6 [58.1–69.0] | 2.7 [−3.4–8.9] *p* = 0.373 |
| | 44/117 | Bone marrow-derived MSCs | 61.5 [57.1–65.9] | −0.3 [−6.0–5.4] *p* = 0.919 |
| | 3/6 | Synovium-derived MSCs | 7.4 [36.7–98.2] | −6.0 [−8.5–20.5] *p* = 0.412 |
| | 11/19 | Adipose-derived stem cells | 56.3 [49.9–62.6] | −5.9 [−11.3−−0.4] ***p* = 0.036** |
| | 8/14 | Bone marrow aspirate | 54.7 [39.8–69.6] | −7.6 [−20.5–5.2] *p* = 0.239 |
| | 3/7 | Bone marrow-derived mononuclear cells | 74.1 [27.9–100.0] | 12.9 [−8.6–34.3] *p* = 0.238 |
| 4. Cell manipulation | 14/27 | During surgery: harvesting, implantation | 58.9 [51.3–66.5] | Surgery vs. Expansion −2.4 [−10.8–5.9] *p* = 0.564 |
| | 59/180 | Expansion: harvesting, expansion *in vitro*, implantation | 61.4 [57.6–65.1] | Surgery vs. Differentiation −4.2 [−13.5–5.1] *p* = 0.374 |
| | 27/58 | Differentiation: harvesting, differentiation *in vitro*, implantation | 63.1 [57.6–68.6] | Expansion vs. Differentiation −1.7 [−8.2–4.7] *p* = 0.594 |

### Cell manipulation

Comparing differences in cartilage regeneration between biomaterials loaded with cells which were not cultured *in vitro* (implanted immediately after harvesting of cells) or were expanded and/or differentiated *in vitro* indicated that cell manipulation did not affect cartilage regeneration (Table 1).

### Sensitivity analyses

To investigate the robustness of the meta-analysis, sensitivity analyses were performed regarding the overall effect of the addition of cells to biomaterials. The overall outcome

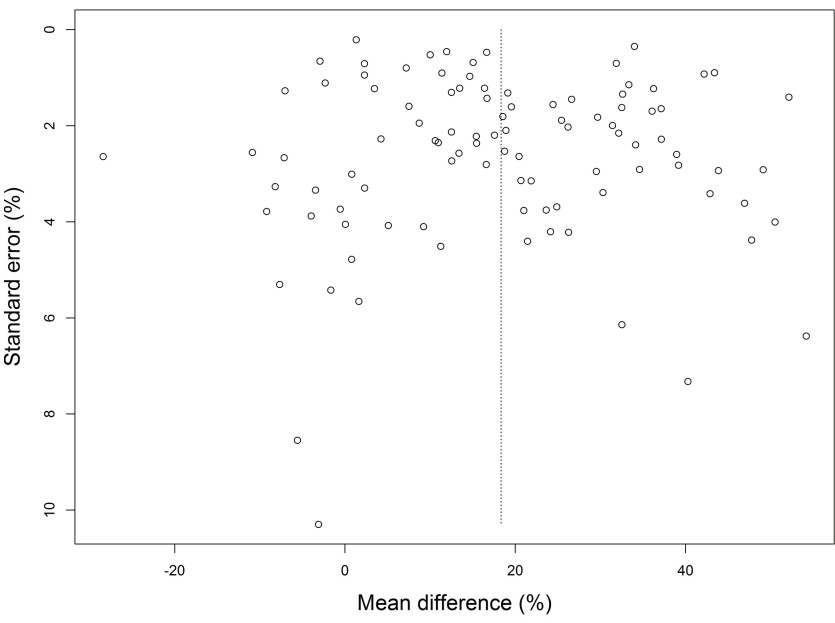

**Figure 4** **Funnel plot of the studies included in the meta-analysis comparing cartilage regeneration using cell-laden and acellular biomaterials.** No substantial asymmetry was found.

effect for cellular scaffolds was not notably affected by the exclusion of studies (1) with assumptions (2) or written in Chinese (no risk of bias assessment performed). Also for acellular biomaterials, the exclusion of these studies had no effect on cartilage regeneration.

### *Publication bias*

Publication bias was assessed for all studies included in the meta-analysis comparing cartilage regeneration using acellular versus cellular biomaterials. Although the funnel plot (Fig. 4) is rectangular in shape, no major asymmetry was observed, giving no indication for publication bias ($p$-value 0.866).

## DISCUSSION

Bone marrow stimulation can be applied to induce cartilage regeneration. Despite therapy, the formed neotissue generally consists of fibrous cartilage, which lacks mechanical and biological properties of native tissue (*Dai et al., 2014*). Therefore, microfracture results in temporary clinical improvement only (*Saris et al., 2014*). To regenerate more durable cartilage tissue, regenerative medicine and tissue engineering may offer a promising addition to bone marrow stimulation by the implantation of scaffolds, which can act as a template to guide and stimulate cartilage regeneration (*Cucchiarini et al., 2014*). In a previous systematic review, the quality of newly formed cartilage in animals was improved by the implantation of biomaterials after bone marrow stimulation, which was further enhanced by loading biomaterials with biologics (*Pot et al., 2016*). The aim of this systematic review was (a) to provide a comprehensive and systematic overview of all current literature regarding animal studies on cartilage regeneration using cellular versus

acellular biomaterials and to identify knowledge gaps, (b) to assess the efficacy of cartilage regeneration using cellular versus acellular biomaterials and to investigate the effect of various parameters (i.e., stem/somatic cells, cell source, cell culture conditions), (c) to gain insight in the methodological quality of animal studies, and (d) to improve the design of future animal models and eventually clinical trials.

In animal studies, the implantation of cellular biomaterials in animal models significantly improved cartilage regeneration by 18.6% compared to acellular biomaterials. Seeding of cells is a major component of the tissue engineering paradigm, which may stimulate healing by the production of many bioactive components. Therefore, the addition of cells to biomaterials enhanced the regenerative process (*Wang et al., 2017*). The heterogeneity ($I^2$) for the main research question and subgroup analyses was very high. Results should therefore be interpreted with caution, especially for subgroup analyses with a limited number of studies. Further clinical studies are required to assess the potential beneficial effect of cellular biomaterials versus acellular biomaterials in patients. *Marcacci et al. (2005)* published promising results of a multicenter clinical phase III retrospective cohort study in which patients were treated with an implant consisting of autologous chondrocytes grown on Hyalograft C, a hyaluronic acid derivative, with a 3-year follow-up. Assessment indicated major clinical improvements and hyaline-like cartilage for the majority of biopsies.

In a subgroup analysis, no significant differences were found between somatic cells and stem cells. Differences were found between various cell types. Adipose-derived stem cells (ADSCs) reduced cartilage regeneration in the subgroup analysis. However, cartilage regeneration using biomaterials seeded with ADSCs was still superior to biomaterials without cells. As compared to other cell types, the origin of ADSCs from fatty tissue may have resulted in significantly reduced cartilage regeneration compared to cells derived from cartilage or subchondral bone. MSCs and chondrocytes have distinct advantages. MSCs are not limited by donor-site morbidity and matrix production after expansion *in vitro* (*Bernhard & Vunjak-Novakovic, 2016*), can be harvested from numerous sources, maintain their multipotency after expansion *in vitro*, can differentiate into chondrocytes that produce cartilage matrix and may suppress proinflammatory cytokines by their immunoregulatory properties. Chondrocytes on the other hand do not terminally differentiate after chondrogenic differentiation, which results in bone formation (*Bernhard & Vunjak-Novakovic, 2016*), and are more easy to manipulate (*Deng et al., 2016*). In clinical trials, the addition of MSCs or chondrocytes to biomaterials resulted in comparable cartilage regeneration (*Nejadnik et al., 2010*; *Lee et al., 2012*). In this study no subgroup analysis was performed to investigate the culture of cell-loaded scaffolds in bioreactors. *Bernhard & Vunjak-Novakovic (2016)* described the beneficial effects of culturing cell-loaded scaffolds in bioreactors with mechanical loading protocols, as these scaffolds more closely resembled the native compressive properties of cartilage tissue and as the applied force steered the location and alignment of cartilage matrix deposition by chondrocytes (*Bernhard & Vunjak-Novakovic, 2016*).

Study characteristics showed a large heterogeneity between studies due to differences in animal model, performed surgery, implanted biomaterial and follow-up period. To reduce the influence of possible confounding parameters, we excluded studies using healthy

animals in which created defects were not filled during the first surgery and osteoarthritis animal models, despite their greater relevance for future applications to treat patients with osteoarthritis.

Various outcome measures were used to investigate cartilage regeneration, including MRI, macroscopic and histological evaluation (more extensively discussed in *Pot et al., 2016*). We selected data from semi-quantitative histological scoring systems as outcome measure, because histological scores are frequently used and allow for quantitative comparisons between studies. However, different scoring systems are available (extensively reviewed by Rutgers et al. (*Hooijmans et al., 2014*)) that assess different processes, e.g., cartilage regeneration only, cartilage and subchondral bone regeneration, and additional biomaterial degradation. Not discriminating between these parameters may be considered as a limitation, but usage of all scoring systems may provide an extensive and complete overview of all aspects affecting the regenerative process. Additionally, evaluation of cartilage regeneration using semi-quantitative histological scoring may still be observer-dependent and subjective, possibly inducing observer (detection) bias. Therefore, it may be better to combine histological scores with biochemical parameters and biomechanical properties, but the ideal combination of outcome parameters remains unknown (*Hooijmans et al., 2014*).

The methodological quality assessment was performed to evaluate the experimental designs and reliability of the results of included studies. The methodological quality (internal validity) is of great importance since a low methodological quality may result in an overestimation or underestimation of the effect size (*Higgins et al., 2011*). No studies were included in or excluded from the meta-analysis based on methodological quality assessment results. Generally, the possibility of assessing the actual risk of bias was limited due to the absence of important details regarding the experimental set-up in most studies and method of randomization. It may be that the animal studies were performed well, but that experimental designs were only reported poorly (*Hooijmans et al., 2012*). For the analysis of the histological sections, however, most studies described that sections were randomized and that outcome assessors were blinded. Detection/observer bias may be introduced in case blinding was not performed and can result in an overestimation of the actual effect of the therapy (*Bello et al., 2014*). The overall validity of the study results may be impaired by bias due to the lack of blinding and randomization (*Bebarta, Luyten & Heard, 2003*; *Hirst et al., 2014*). Reporting of animal studies may be improved by using standardized protocols, including the ARRIVE guidelines (*Kilkenny et al., 2012*) or golden standard publication checklist (*Hooijmans et al., 2011*).

A high translational value of animal studies is crucial to take treatments forward to clinical practice. Therefore, validated and predictive animal models are required. Many challenges and limitations are associated with the use of animal models for cartilage defects. *Chu, Szczodry & Bruno (2010)* and *Ahern et al. (2009)* extensively described strengths and shortcomings of different animal models related to e.g., joint size, cartilage thickness, defect size, intrinsic healing potential and animal maturity, in comparison to lesions in clinical studies. In most animal experiments, the follow-up period was maximally six months, while in patients clinical improvements are generally observed up to 1.5–3

years after microfracture surgery (*Hoemann et al., 2010*; *Van der Linden et al., 2013*). The translational value and considerations to select animal models were extensively discussed before (*Pot et al., 2016*).

Improved reporting of animal studies is required in future studies and studies should strive to resemble the clinical situation to facilitate translation. For clinical application of new regenerative medicine and tissue engineering strategies, including the use of biomaterials, biologics and cells, the effectiveness needs to be proven both in animal models and clinical studies (*Cousin et al., 2016*). Moreover, the cost-effectiveness of new interventions in clinical practice may be assessed using early health economic models (*De Windt et al., in press*). Considerations for the addition of cells to biomaterials are of great importance and limitations (including donor-site morbidity, cell culture costs, regulatory issues, limited off the shelf availability, and potential multiple-stage surgical procedures (*Pot et al., 2016*; *Efe et al., 2012*)) should be weighed against potentially superior cartilage regeneration by applying cellular biomaterials. Difficulties in controlling cell culture and the development of novel materials stimulating tissue regeneration may justify the use of acellular biomaterials. Future research focusing on biomaterials properties, source and manipulation of cells, and possibly patient profiling, may allow selection of the best treatment for each individual patient (*Kon et al., 2015*).

## CONCLUSION

This systematic review and meta-analysis provides an extensive overview of all animal studies applying regenerative medicine and tissue engineering approaches to regenerate articular cartilage by implantation of cellular versus acellular biomaterials after applying bone barrow stimulation. Cartilage regeneration was more effective by implantation of cellular biomaterials compared to acellular biomaterials. This study together with a previous study on the beneficial properties of scaffolds and growth factors implies that all components of the tissue engineering paradigm can be valuable for improved regeneration of articular cartilage.

## ACKNOWLEDGEMENTS

We thank Jie An (Department of Biomaterials, Radboud Institute for Molecular Life Sciences, Radboud university medical center) and Chunling Tang (Department of Tumor Immunology, Radboud Institute for Molecular Life Sciences, Radboud university medical center) for their contribution to the paper. Gerrie Hermkens from the Radboud university medical center library is greatly acknowledged for help retrieving full-text studies.

### Funding

This work was supported by a grant from the Dutch government to the Netherlands Institute for Regenerative Medicine (NIRM, grant No. FES0908). Rob de Vries received funding from The Netherlands Organisation for Health Research and Development (ZonMw; grant

nr. 104024065). The sources of funding have no other involvement in this publication. The funders had no role in study design, data collection and analysis, decision to publish, or preparation of the manuscript.

## Grant Disclosures

The following grant information was disclosed by the authors:
Netherlands Institute for Regenerative Medicine: FES0908.
Netherlands Organisation for Health Research and Development: 104024065.

## Competing Interests

The authors declare there are no competing interests.

## Author Contributions

- Michiel W. Pot conceived and designed the experiments, performed the experiments, analyzed the data, contributed reagents/materials/analysis tools, wrote the paper, prepared figures and/or tables, reviewed drafts of the paper.
- Toin H. van Kuppevelt conceived and designed the experiments, reviewed drafts of the paper.
- Veronica K. Gonzales performed the experiments, reviewed drafts of the paper.
- Pieter Buma reviewed drafts of the paper.
- Joanna IntHout analyzed the data, contributed reagents/materials/analysis tools, prepared figures and/or tables, reviewed drafts of the paper.
- Rob B.M. de Vries conceived and designed the experiments, analyzed the data, contributed reagents/materials/analysis tools, reviewed drafts of the paper.
- Willeke F. Daamen conceived and designed the experiments, analyzed the data, reviewed drafts of the paper.

## Data Availability

   The raw data is included in Supplemental Information 6.

## Supplemental Information

Supplemental information for this article can be found online at http://dx.doi.org/10.7717/peerj.3927#supplemental-information.

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
