# Peer review of "Augmented cartilage regeneration by implantation of cellular versus acellular implants after bone marrow stimulation: a systematic review and meta-analysis of animal studies"

_PeerJ, doi:10.7717/peerj.3927_

## Round 0.1 · original submission · Major Revisions

This manuscript is an interesting analysis of published literature and a good review on cartilage regeneration. However, there are many concerns raised by all the three reviewers. If authors can revise their manuscript according to reviewer's suggestions and provide point by point reply to the reviewer's comments by including needed information and modifying their manuscript, it would be helpful for the PeerJ readers to appreciate their review very much.

·

Basic reporting

'no comment'

Experimental design

'no comment'

Validity of the findings

'no comment'

Additional comments

Manuscript entitled “Augmented cartilage regeneration by implantation of cellular versus acellular implants after bone marrow stimulation: a systematic review and meta-analysis of animal studies (#18289)” is well written and discussed. However, following comments should take care during the revision.
1. Manuscript lacking the future implication of this kind of analysis in cartilage tissue engineering. Please discuss the importance of the study in discussion and conclusion parts of the manuscript.
2. Growth factors are the one of the important components of tissue regeneration along with scaffold and cells. It would be nice to include the growth factors description in the manuscript.
3. Stimulation in form of bioreactor is again one of the crucial factors for differentiation of stem cells into chondrocyte lineage. Please also include the bioreactor in search, and discussion in the manuscript.
4. Recently, cryogels special kind of the hydrogel which synthesized at sub-zero temperature to achieve good porosity and mechanical strength in the scaffold, gained significant attention in cartilage tissue regeneration. Please also include cryogels (PMID:28230077, 27886065, 27185069, 25498693 etc.) in the search study.
5. In line 185, what’s I2? Please define and discuss the importance of I2.
6. Line 132, what’s Safranin-O-staining does in cartilage research? Please discuss briefly.
7. Please briefly discuss “Cochrance approach” in line 153.
8. Conclusion lacking the importance of the study.
9. Please include growth factors and bioreactor in figure 1 illustration of cartilage tissue engineering.

·

Basic reporting

The authors have used clear language throughout and present a well defined research question with good background information.

I recommend removing the word "unbiased" from line 80 and line 312. A systematic review is not an unbiased overview as it is subjected to biases at all stages of the review process. For example, the authors make decisions on types of study to include, which databases to search and the methods to use.

Line 284-287. Please include effect size data for adipose-derived stem cells (mean and confidence intervals).

Experimental design

This is a primary research article with a well defined research question- to present a overview of all studies reporting regeneration of articular cartilage by implantation of cell-laden versus cell-free biomaterials in the knee and ankle joint after bone marrow simulation in animals comparing different cell types and methods.

Did the authors publish a study protocol before carrying out their review? Can you provide a link to this protocol or a copy of the protocol?

I would recommend that the authors include the specific search terms used in their search strategy instead of referencing other publications. This could be included in the methods or as supplementary information.

I would question whether the use of semi-quantitative histological scoring systems (which are observer-dependent and thus subjective (Ref. Rutgers et al. 2010) is the best method to determine the success of regeneration? I suggest discussing the limitations of this as outcome measure in the review.

I do not understand the meta-analysis methods used specifically, line 162-163. "modeled with a compound symmetry covariance matrix, as this resulted in the lowest Akaike Information Criterion value." Can you give more details?

I would recommend statistically testing for asymmetry in the funnel plot.

The reviewers do not report data extraction being carried out by two independent reviewers. I suggest discussing the limitations of this.

Validity of the findings

The included data has appropriate controls.

As requested above, I would like a bit more information on the meta-analysis methods before assessing if the results are statistically sound.

The conclusions based on the results appear to be justified and linked to the original research question.

Additional comments

This is a well written review. I would recommend that the authors revise the manuscript bu clarifying some methods.

Reviewer 3 ·

Basic reporting

The authors of this manuscript have presented a summarized overview and meta-analyses on the previously published research articles on the augmented cartilage regeneration after bone marrow stimulation. This study utilized various tools for screening the suitable reports on this topic with full details of experiments to compare the results for concluding remarks. This review suggests that cellular biomaterials/implants have higher cartilage regeneration efficacy compared to acellular biomaterials/implants. In my understanding, this study of authors is the continuation of their previous finding (Ref. 8 of the manuscript), which demonstrated the role of acellular biomaterials in cartilage regeneration. The manuscript is written well, however, few grammatical errors have been noticed, that can be rectified in revision.

Experimental design

This is a systematic review and meta-analysis of the already published research. The findings are relevant to understand the suitable combination of biomaterials, biologics, and cells for the augmented cartilage regeneration. The study was started with search finding of 11,248 reference articles on this topic, which were further screened rigorously to obtain 146 full-text studies for further analysis.

Validity of the findings

The provided methods of research data screening and meta-analyses are suitable to demonstrate the unbiased conclusion on the role of cellular/acellular biomaterials in cartilage regeneration after bone marrow stimulation. Authors of this study concluded that cellular biomaterials have shown 18.6% higher regeneration capacity compared to acellular biomaterials.
I believe that the provided hypothesis on augmented cartilage regeneration using cell seeded biomaterials after bone marrow stimulation is not significant to understand the basics behind higher regeneration ability. I may suggest discussing more in this aspect in the discussion section using suitable reference articles will enhance the understanding and readership quality.

Additional comments

I have found few points that need to take care by authors of this manuscript;
1. Section 2.1_the provided ref. (like 32, 33..) are not clear. Can authors justify the relevance of these references? If these references are used in support of methods used, then they need to be cited with proper elaboration.
2. section 2.2, 2.4 etc., Authors have used the term 'reviewer' at few places to demonstrate their role, I believe that the correct terminology to indicate their role by the name of 'authors' of the present study.
3. I have found the repetition of many sentences in little-modulated form, for example; line 228-230 'cells were either.....' is providing the same meaning as provided in the next sentence, line 230-233, 'in some studies....'. Authors should revise the manuscript in this aspect to make more crisp and attractive text content.

---

## Round 0.2 · accepted · Accept

Dear Michiel, your revised manuscript is now acceptable for publication in PeerJ.

Thank you for your submission and revision.

·

Basic reporting

Well-defined question, clearly written

Experimental design

Methods and analysis described clearly and included sufficient details.

Validity of the findings

The conclusion is well stated and directional for preclinical research.

Additional comments

Thanks to the authors for addressing my comments. The manuscript is significantly improved and readable well. Now, I am recommending the manuscript for publication in PeerJ journal.

Reviewer 3 ·

Basic reporting

NA

Experimental design

NA

Validity of the findings

NA

Additional comments

The modified manuscript is showing improvement in its technical information and better readership quality. The revised manuscript is now recommended for consideration for publication in Peer J.